# Challenges and Opportunities in Identifying and Characterising Keratinases for Value-Added Peptide Production

**Juan Pinheiro De Oliveira Martinez [1], Guiqin Cai [1], Matthias Nachtschatt [1], Laura Navone [1], Zhanying Zhang [1], Karen Robins [1,2] and Robert Speight [1,*]**

[1]   Science and Engineering Faculty, Queensland University of Technology, Brisbane, QLD 4000, Australia; j5.martinez@hdr.qut.edu.au (J.P.D.O.M.); guiqin.cai@qut.edu.au (G.C.); m.nachtschatt@qut.edu.au (M.N.); laura.navone@qut.edu.au (L.N.); jan.zhang@qut.edu.au (Z.Z.); sustainbiotech@iinet.net.au (K.R.)

[2]   Sustain Biotech, Sydney, NSW 2224, Australia

[*]   Correspondence: robert.speight@qut.edu.au; Tel.: +61-7-3138-0373

**Abstract:** Keratins are important structural proteins produced by mammals, birds and reptiles. Keratins usually act as a protective barrier or a mechanical support. Millions of tonnes of keratin wastes and low value co-products are generated every year in the poultry, meat processing, leather and wool industries. Keratinases are proteases able to breakdown keratin providing a unique opportunity of hydrolysing keratin materials like mammalian hair, wool and feathers under mild conditions. These mild conditions ameliorate the problem of unwanted amino acid modification that usually occurs with thermochemical alternatives. Keratinase hydrolysis addresses the waste problem by producing valuable peptide mixes. Identifying keratinases is an inherent problem associated with the search for new enzymes due to the challenge of predicting protease substrate specificity. Here, we present a comprehensive review of twenty sequenced peptidases with keratinolytic activity from the serine protease and metalloprotease families. The review compares their biochemical activities and highlights the difficulties associated with the interpretation of these data. Potential applications of keratinases and keratin hydrolysates generated with these enzymes are also discussed. The review concludes with a critical discussion of the need for standardized assays and increased number of sequenced keratinases, which would allow a meaningful comparison of the biochemical traits, phylogeny and keratinase sequences. This deeper understanding would facilitate the search of the vast peptidase family sequence space for novel keratinases with industrial potential.

**Keywords:** keratinase; serine protease; metalloprotease; peptidase; keratin hydrolysis; keratin waste; valorisation; bioactive peptides

## 1. Introduction

Millions of tonnes of waste keratin are produced every year in the poultry, meat processing, leather and wool textile industries. The global poultry meat processing industry alone produces $40 \times 10^6$ tonnes of waste feathers annually [1]. With the transition away from the fossil fuel-centric economy to a sustainable circular economy, the valorisation of keratin materials addresses the waste problem and facilitates the integration of waste keratin into new value chains to enable a circular economy.

Traditionally, keratin waste has been sent to landfill or rendering, or used as fertilizer, feather meal or incinerated [2,3]. There is, however, an opportunity for livestock industries to produce higher value products from waste keratin. There are multiple thermochemical methods available to prepare hydrolysed keratin for various value-adding opportunities [4]. However, the use of peptidases

with keratinolytic activity for keratin hydrolysis protects the integrity of the keratin amino acids in most cases and allows control over the peptide size in the hydrolysate that is not readily achievable with other methods [5]. This degree of control allows the production of bespoke medical biomaterials, smart biocomposites, protein feed supplements with enhanced nutritional and bioactive properties as well as personal care products with enhanced functional and bioactive properties.

Identifying peptidases with keratinolytic activity is an inherent problem associated with the search for new enzymes. Keratinase activity however appears to be dependent on the accessibility of the keratin substrate to the enzyme [6,7]. Thermochemical or biochemical treatment of the keratin, with emphasis on the reduction of the disulphide bond and disruption of other important bonds involved in the structural stability of keratin like isopeptide, hydrogen and glycolytic bonds [6,8–10], appears to be the prerequisite for enzymatic hydrolysis. Sulphitolysis, which involves reduction of the disulphide bond in keratin, often acts synergistically with keratinases in nature [6,7]. Although destabilization of the keratin structure is a prerequisite for keratin hydrolysis, not all peptidases can hydrolyse keratin. Peptidases like trypsin, papain and pepsin cannot hydrolyse keratin as efficiently as peptidases with keratinolytic activity, even if the reduction of disulphide bond has already occurred [11]. The elucidation of the unique characteristics of peptidases with keratinolytic activity that differentiate them from the other peptidases, would be an important breakthrough in the search for new and robust keratinases for the valorisation of keratin waste.

This paper reviews twenty sequenced peptidases with keratinolytic activity from the serine protease and metalloprotease families by comparing their biochemical characteristics and will highlight the difficulties associated with the interpretation of these data.

## 2. Keratin: A Complex and Strong Structure

Keratins are important structural proteins produced by vertebrate epithelia that have various physiological function. Keratins can act as a protective barrier to water, against infection or cushion tissue from mechanical impact. The two main types of keratins proteins are α-keratin and β-keratins. These two types are further divided into acidic or basic, soft or hard, and have different molecular weights [4,9,12,13]. The following section describes the complexity of the keratin structure, which provides insight into the resistance of keratin to hydrolysis. This review will concentrate on hard α-keratin and β-keratin, γ-keratins and the keratin-associated proteins, which are common to mammalian hair, bristles, wool, hooves, horns and feathers.

α-Keratin has an α-helix structure, which is stabilized by hydrogen bonding and the presence of multiple cysteines forming disulphide bridges. α-Keratin is characterized by a lower sulphur content compared to other keratins and a molecular mass of 60–80 kDa [4]. Hard α-keratin is the major protein of mammalian fibres, nails, hooves and horns. In contrast, hard β-keratins are characteristic of the hard, cornified epidermis of reptiles and birds, e.g., feathers, claws and scales, and have a twisted β-sheet-like structure. They also form the major component of the fibre cuticle. The β-keratin pleated sheets consist of β-strands, which are laterally packed and can have a parallel or antiparallel orientation. The β-sheets are held together by hydrogen bonds and the planar nature of the peptide bond, which results in the stable pleated β-sheet [13]. β-Keratins have a molecular mass of 10–22 kDa. A third type of keratin, γ-keratin, is a globular protein with a high sulphur content and a molecular weight of about 15 kDa. This keratin, along with keratin-associated proteins, form the matrix between the microfibrils and microfibrils of the fibre cortex of mammalian fibres and stabilize the structure of the cortex via extensive disulphide bridge formation.

The complex structural organization of all mammalian fibres is very similar [8]. The hair fibre consists of an outermost cuticle layer, which is composed of overlapping flattened scale-like cells that form a protective sheath around the cortex [8]. The major protein of the fibre cuticle is β-keratin [4]. The cortex is composed of hard α-keratin intermediate filaments embedded in a sulphur-rich matrix. These filaments surround the medulla when present, as is the case for coarser fibres. The cell membrane complex binds the cuticle and cortical cells.

The cuticle layer is laminated and consists of the following layers—the cuticle filament-associated surface membrane, the cystine-rich exocuticular *a*-layer, the lower exocuticle and the endocuticle, which contains only low levels of sulphur-containing amino acids and constitutes the inner lining of the cuticle [8]. The outermost layer of the cuticle provides a hydrophobic barrier, which protects the fibre surface from water and chemical compounds. This cuticle filament-associated surface membrane is 2–7 nm thick and composed of highly cross-linked proteins and lipids. The major fatty acid of the cuticle surface lipids found in human and animal hair is 18-methyleicosanoic acid [14]. It is covalently linked to the protein matrix below by a thioester linkage and the protein matrix is cross-linked by isopeptide bonds [15]. An isopeptide bond results from the transglutaminase-catalysed formation of an amide bond between the amino acid side chains of the amino acid residues in the keratin protein, for example, lysine and glutamine [9].

The cortical cells are assembled as keratin intermediate filaments and have a diameter of 7–8 nm in all mammalian fibres [8]. These intermediate filaments form ordered aggregates or microfibrils and macrofibrils depending on species and function (Figure 1). The hard α-keratin intermediate filaments are assembled from tetramers, a pair of laterally aligned and antiparallel dimeric molecules. On average, keratin intermediate filaments contain eight tetramers. In the case of wool, the cortex region is composed of an orthocortex and paracortex with different intermediate filament/matrix packing. The proportion of ortho- and paracortex in the wool fibre determines the degree of crimping [13].

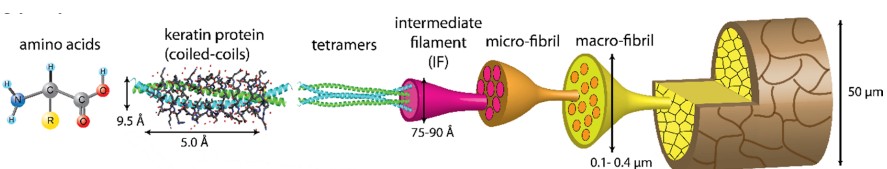

**Figure 1.** Structure of keratin. Adapted from work in [12] under the Creative Commons Attribution 4.0 International license (https://creativecommons.org/licenses/by/4.0/deed.en).

Keratin peptide heterodimers are formed when a type I (acidic) polypeptide chain and a type II (basic) polypeptide chain align in parallel. Each polypeptide chain is composed of a central α-helical region (about 46 nm in length) with non-helical head and tail domains [13]. The head and tail domains are rich in cysteine, glycine and tyrosine amino acids. Disulphide and isopeptide bonds are formed with other keratin intermediate filaments, cysteine-rich matrix proteins and keratin associated proteins, which stabilize the fibre [8–10]. The disulphide bonds along with the N-acetyl glucosamine-glycosylated serine and threonine in the head and tail domains also stabilize the heterodimers [6].

## 3. Thermochemical Methods of Keratin Degradation

There are multiple thermochemical methods available to prepare hydrolysed keratin for various value-adding opportunities, with specific processes chosen depending on the end-use [4]. Thermochemical methods include solubilization of keratin in organic solvents, ionic liquids or by hydrothermal methods; oxidation or reduction of the disulphide bridges; disruption of the hydrogen bonds with compounds like urea; and acid or base hydrolysis.

The composition of the final hydrolysate will depend on the method used to hydrolyse the keratin. Some of the thermochemical processes result in a hydrolysate containing a highly diverse mix of keratin-derived peptides and free amino acids and others are more specific. However, in most cases, the amino acid composition is modified. The processes and hydrolysate products will be described in more detail in the following section.

After solubilization of keratin with solvents like with *N,N*-dimethylformamide or dimethyl sulfoxide, precipitation is required with acetone and drying to produce a powder of keratin [16]. The major drawback of this method is the use of large quantities of solvents, which need to be recycled or incinerated. Solubilization can also be achieved with ionic liquids. Xie et al. used the ionic liquid, 1-butyl-3-methylimidazolium chloride for the solubilization of wool keratin, which disrupted

the hydrogen bonds in the keratin macromolecules [17]. The keratin peptides were precipitated from the resulting hydrolysate with methanol. Ionic liquids are more expensive than traditional solvents and extraction of the keratins from the ionic liquid can be difficult.

Hydrothermal treatment is usually carried out at temperatures of 80–140 °C and steam pressures of 10–15 psi. Acid or base can be added to speed up the process of solubilization [18]. Under conditions of high temperature and pressure, the thermally unstable amino acids, glutamine and asparagine are degraded [19]. If base is added to this process then lysine, methionine and tryptophan are also destroyed [20,21]. Modified amino acids, lysinoalanine and lanthionine are also formed from lysine and cystine, respectively. Heating of proteins leads to a degree of racemization of the free and bound L-amino acids [22–24].

Reduction with reducing agents like thioglycolate [4], dithiothreitol [25], 2-mercaptoethanol [26], sodium sulphite [27], bisulphites [28] or cysteine [29] combined with high concentrations of compounds like urea, thiourea or surfactants, which disrupt the hydrogen bonds stabilising the keratin structure, results in the production of kerateine [30]. Kerateine contains cysteine thiol and cysteine sulfonate in place of the disulphide bonds. Kerateine is less soluble in water and can be re-cross-linked if exposed to an oxidant [4].

The microstructure of wool keratin after treatment for 4 h at 65 °C with 2-mercaptoethanol, EDTA, high concentrations of urea and pH 9 was investigated by Cardamone [31]. Analysis of the hydrolysate revealed a defined mixture of microfibrillar and intermediate filaments. This mixture of subunits was suitable for producing self-assembling biomaterials.

Oxidation of keratin by oxidants like peracetic acid [32] or peroxycarboximidic acid [33] leads to the formation of keratose. Keratose contains sulfonic acid groups and cysteic acid instead of the disulphide bonds [4]. These keratoses are hydroscopic, water soluble and the disulphide bridges cannot spontaneously re-form under oxidative conditions. Keratoses are not as stable as kerateines.

Oxidative sulphitolysis has been patented and commercialized to produce three functional keratin protein and peptide products. These products are based on S-sulphonated keratin intermediate filaments, S-sulphonated keratin high-sulphur proteins and keratin peptides [34]. The process aims at maintaining the structural integrity of the keratin proteins. The cystine groups in the wool keratin are converted to S-sulfocysteine using sodium sulphite or sodium metabisulfite and then oxidized with cupraammonium hydroxide. The intermediate filaments and peptides can undergo crosslinking by reductive desulfonation of the cysteines in the filaments and peptides and subsequent reformation of the intermolecular disulphide bonds.

One of the disadvantages of alkaline hydrolysis of keratins is the modification or degradation of amino acids (Table 1). Alkaline hydrolysis of keratins at higher temperatures results in the degradation of the thermally unstable amino acids, asparagine, glutamine, arginine, serine, threonine and cysteine [5]. Lysinoalanine and 8-aminoalanine are formed under alkaline conditions [35,36]. Another modification that occurs is the racemization of free or bound L-amino acids to the D-enantiomers [23,37,38]. Free amino acids racemize ten times slower than bound amino acids [24]. Following, for example, prolonged treatment of wool keratin at 70 °C and pH 9–11, lanthionyl residues [31] and dehydroalanine [39] are formed from cystine. Cystine and hydroxy amino acids were destroyed if the alkaline treatment was performed in the presence of reducing agents [40].

**Table 1.** Amino acid modification during alkaline treatment.

| Amino Acid | Degradation Products | Reference |
|---|---|---|
| Asparagine | Aspartate, ammonia | [41] |
| Glutamine | Glutamate, ammonia | [41] |
| Arginine | Ornithine, citrulline, 3-aminopiperidin-2-one | [42] |
| Serine | Glycine, alanine, oxalic acid, lactic acid, ammonia | [43] |
| Threonine | Glycine, alanine, α-aminobutyric acid, ammonia | [44] |
| Cysteine | Pyruvic acid, sodium sulfide, ammonia | [45] |
| * Cystine, lysine, arginine | Lanthionine, lysinoalanine, ornithinalanine; * dehydroalanine [39] | [35,36] |
| L-amino acids | D-amino acids | [37] |

Note: * Dehydroalanine is probably formed from the cleavage of the C-S bond in cystine.

Acid hydrolysis of keratins leads to the loss of some amino acids like serine, threonine, tyrosine and cystine and the conversion of asparagine, glutamine, methionine and tryptophan into other compounds ([5,19,46] Table 2). Polypeptides, resulting from the acid hydrolysis of keratin, have a more amorphous structure than alkaline hydrolysates, because most of the hydrogen bonds are broken during this process [47]. A typical acid hydrolysis of keratin uses hydrochloric acid [48,49] or sulphuric acid [50] at high temperatures.

Zhang et al. showed that acid hydrolysis was not as effective as other treatments mentioned above [49]. Wool keratin was hydrolysed with 4M hydrochloric acid at 95 °C for 24 h, resulting in 33% solubilization of the wool keratin. Increasing the treatment time had no effect on the yield, suggesting that there is a recalcitrant portion of the keratin resistant to acid hydrolysis.

Thermochemical methods offer cheap and versatile processes for hydrolysing keratin for a variety of applications. However, the use of harsh chemicals and conditions, the lack of ability to control the processes in most cases and the often unfavourable modification of the amino acids or peptides present environmental problems and peptide mixes that would be unsuitable for some applications. Using enzymes working under mild conditions to catalyse the hydrolysis offers a favourable alternative.

**Table 2.** Amino acid modification during acid treatment.

| Amino Acid | Degradation Products | Reference |
|---|---|---|
| Asparagine | Aspartate, ammonia | [19,41] |
| Glutamine | Glutamate, ammonia | [19,41] |
| Methionine | Methionine sulfoxide | [19] |
| Tryptophan | Oxindolylalanine, dioxindolylalanine | [46] |

## 4. Microbial Degradation of Keratin

The first peptidases with keratinolytic activity were found in *Bacillus* sp. and *Streptomyces* sp. and belong to the serine peptidase family [51]. The ability to degrade keratin is widespread and has been identified in bacteria and fungi [4,52]. Keratin-degrading microorganisms have been isolated from many sources like skin, feathers, hair, nails, soil, geothermal hot stream and wastewater, which is reflected in the optimum pH and temperature of the keratinase activity of these microorganisms. The pH optimums of keratinases range from pH 5.5 for the fungal keratinase from *Trichophyton mentagrophytes* [53] to pH 12.5 for the keratinase from *Brevibacillus* sp. AS-S10-11 [54]. Although, temperature optimums vary from 30 °C for the keratinase from *Brevibacterium luteolum* [55] to 100 °C for the keratinase from *Fervidobacterium islandicum* AW-1 [56].

Publications from 2018 and 2019 report the isolation of diverse species of bacteria like *Streptomyces* sp. [57], *Aeromonas hydrophila* FB3 [58], *Pseudomonas putida* KT2440 [59] and *Serratia marcescens* EGD-HP20 [60,61] with keratinolytic activity. However, the number of *Bacillus* strains with keratinolytic activity prevailed over any other genus of bacteria [62–86]. Valorisation of waste feathers [5,65,66,87] and the replacement of the traditional, highly polluting hide dehairing step used in the leather industry with a more environmentally friendly enzymatic step using keratinases [2,55,79,88] were the dominant themes of these papers.

Despite the interest in the enzymatic hydrolysis of keratin, mechanisms of keratin degradation in microorganisms are not fully understood. There is evidence that microbial degradation of keratin proceeds via a consortium of enzymes (Figure 2 [6,7,89]).

Disruption of the keratin structure is an essential step in the breakdown of keratin by keratinases. Various mechanisms have been suggested for fungal systems. Disulphide bond reductases and the intracellular cysteine dioxygenase can break the structure-stabilizing disulphide bridges in keratin [6,7,90]. Cysteine dioxygenase in conjunction with aspartate aminotransferase produces the reducing agent, sulphite, from cysteine, which is secreted into the surroundings and contributes to the chemical reduction of the disulphide bond. The reduction of the disulphide bonds aids access of the endoproteases (serine protease family), exoproteases (metalloprotease family) and

oligopeptidase (metalloprotease family) to the keratin fibres or feathers. It has also been found that the membrane-bound redox system of the cell can cleave the disulphide bonds in keratin. The mechanical pressure exerted by fungal mycelia penetrating the keratin structure can also contribute to the disruption of this structure, facilitating access of the keratinase to the substrate. In nature, these mechanisms act synergistically with keratinases and speed up the degradation of keratin. Auxiliary proteins, like lytic polysaccharide monooxygenases (LPMOs), have been found associated with keratin degradation [6]. It is thought that they contribute to α- and β-keratin degradation. Until now LPMOs were thought to be associated with cellulose, chitin, hemicellulose and starch degradation only. It is possible that these enzymes hydrolyse the glycolytic bond between N-acetylglucosamine and serine and threonine in the head and tail region of the intermediate filaments, which contributes to the destabilization of the keratin structure.

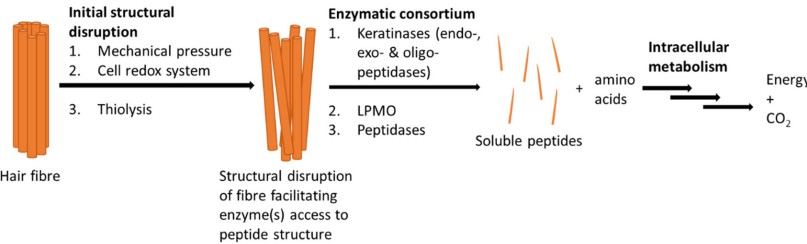

**Figure 2.** Possible mechanisms for microbial degradation of keratin (LPMO = lytic polysaccharide monooxygenase).

However, examples of peptidases with keratinolytic activity that do not need the assistance of disulphide reducing enzymes or agents have also been reported. Pillai et al. isolated a serine protease from *Bacillus subtilis* P13 with reductase and keratinase activities [91]. The isolated enzyme was able to decompose feathers and dehair hides.

He et al. analysed the enzyme consortium involved in the hydrolysis of feathers by a specific strain of *Bacillus subtilis* and identified four of the enzymes involved in keratin hydrolysis [74]: a serine protease with keratinase and disulphide bond-reducing activity; a peptidase T; a γ-glutamyltransferase, which generates a free cysteinyl group from glutathionine; and a cystathionine γ-synthase, which catalyses the production of L-cystathionine from homoserine ester and cysteine. The L-cystathionine is further converted to methionine and ammonia is released.

## 5. Characterisation and Comparison of Keratinases from S1, S8 and M4 Peptidase Families

Many articles characterising organisms capable of degrading keratin and their possible industrial applications have been published. Yet, there are few articles that report enzyme sequences and investigate the molecular and biochemical characteristics of the enzymes produced by these organisms [92,93]. The first paper that explored the molecular aspects of a keratinase produced by *Bacillus licheniformis* was published by Lin et al., 1995. Since then, more than 40 keratinases have been sequenced. To date, peptidases with keratinolytic activity from six different peptidase families have been identified: S1, S8, M4, M5, M14 and M28. Most of the characterized keratinases are produced by *Bacilli* and are members of the S8 serine peptidase family. There are currently over 127,000 peptidase sequences from the S1 (70919), S8 (38270), M4 (6403), M5 (145), M14 (11202) and M28 (904) families deposited on the MEROPS peptidases database. These 127,000 peptidase sequences represent an enormous unmined potential for the discovery of new peptidases with keratinolytic activity if the requisite properties of a peptidase with keratinolytic activity can be identified.

The S1 family sequences, when pairwise aligned, show a minimum value of 27.27% and a maximum of 97.22% identity, with an average of 61.48% for the four available sequences. The S8 family has a minimum of 13.69% and a maximum of 99.72% identity, with an average of 63.29% for the 13 available sequences, and the M4 family has 25.56% identity between the two available sequences. Although many of the characterized enzymes have been produced by the native unmodified organism [94–98], several

examples involve heterologous expression. Different organisms have been used for recombinant production, including yeast such as *Komagataella Pastoris* (*Pichia Pastoris*) [99] and bacteria such as *Escherichia coli* [100–110] and *Streptomyces lividans* [111]. Including the pre-pro-domains with the catalytic domain in heterologous systems have been shown to maintain enzyme activity and secretion [99,102,107,110] and inclusion of C-domains, when present, is important for substrate binding and recognition [105]. Replacing the native signal peptide for the *E. coli* signal peptide when expressing in *E. coli* has also led to higher levels of expression [101].

In this section the biochemical data of twenty sequenced peptidases with keratinolytic activity from the S1 and S8 peptidase families (serine proteases) and the M4 peptidase family (metalloprotease) are compared (Table 3). Difficulties associated with the interpretation of these data are also highlighted. The selection is based on the availability of sequence and biochemical data. The M5, M14 and M28 peptidase families were excluded because each family had only one biochemically characterized example with full sequence data available.

**Table 3.** Keratinolytic microorganisms and their keratinases from the S1, S8 and M4 keratinases selected for this study.

| Organism | Strain | Keratinase Name | Accession No. [1] | Reference |
|---|---|---|---|---|
| **S1A Peptidases** | | | | |
| *Actinomadura viridilutea* | DZ50 | KERDZ | KU550701 | [94] |
| *Actinomadura keratinilytica* | Cpt29 | KERAK-29 | ASU91959 | [95] |
| *Streptomyces fradiae* | Var. k11 | SFP2 | AJ784940 | [99] |
| *Nocardiopsis sp.* | TOA-1 | NAPase | AY151208 | [111] |
| **S8A Peptidases** | | | | |
| *Bacillus circulans* | DZ100 | SAPDZ | AGN91700 | [100] |
| *Bacillus licheniformis* | RPk | KerRP | EU502844 | [96] |
| *Stenotrophomonas maltophilia* | BBE11-1 | KerSMD | KC814180 | [101] |
| *Stenotrophomonas maltophilia* | BBE11-1 | KerSMF | KC763971 | [101] |
| *Bacillus pumilus* | A1 | KerA1 | ACM47735 | [97] |
| *Bacillus pumilus* | CBS | SAPB | CAO03040 | [102] |
| *Bacillus pumilus* | KS12 | rK$_{27}$ | HM219183 | [103] |
| *Bacillus tequilensis* | Q7 | KerQ7 | AKN20219 | [104] |
| *Bacillus cereus* | DCUW | Vpr | ACC94305 | [105,112] |
| *Bacillus altitudinis* | RBDV1 | KBALT | APZ77034 | [63] |
| *Thermoactinomyces sp.* | YT06 | YT06 Protease | WP_037995056 | [98] |
| *Thermoactinomyces sp.* | CDF | Protease C2 | ADD51544 | [106] |
| *Meiothermus taiwanensis* | WR-220 | rMtaKer | 5WSL | [107] |
| *Brevibacillus sp.* | WF146 | WF146 Protease | AAQ82911 | [108] |
| **M4 Peptidases** | | | | |
| *Geobacillus stearothermophilus* | AD-11 | RecGEOker | KJ783444 | [109] |
| *Pseudomonas aeruginosa* | KS-1 | KerP | HM452163 | [110] |

Note: [1] NCBI GenBank nucleotide accession number.

### 5.1. S1, S8 and M4 Peptidase Families

The S1 family is the largest family of serine proteases. The active site of S1 peptidases contains the catalytic triad, His, Asp and Ser. All enzymes characterized in this family are endopeptidases. The four peptidases in Table 3 belong to the S1A family represented by chymotrypsin as the type-example. The hydrophobic amino acid at the P1 site determines the specificity of these peptidases [113,114].

The S8 family is currently the second largest serine protease family and the most widely characterized to date [114,115]. Most of the keratinases are found in the subfamily S8A including the 14 keratinases in Table 3. They are represented by subtilisin as the type-example. Their active site contains the catalytic triad of Asp, His and Ser. In general, these enzymes are endopeptidases [116], active between neutral and moderately alkaline pH and many are thermostable [117]. Most enzymes in this family are not specific, usually cleaving after a hydrophobic residue in the peptide substrate [114,117]. S1 and S8 families are examples of convergent evolution as they catalyse the same reaction but have no sequence homology. Two calcium-binding sites contribute to thermal stability in many members of these families [114,117].

Two keratinases in Table 3 belong to the M4 family. They are characterized by a catalytic zinc ion tetrahedrally coordinated in the active site by a histidine and glutamate present in a HEXXH motif, another glutamate residue and water [118]. Most members of this family are endopeptidases and active at neutral pH. The preferred cleavage site occurs at a hydrophobic residue followed by leucine, phenylalanine, isoleucine or valine. These peptidases are stabilized by $Ca^{2+}$ [119].

Independent of their families, keratinases usually cleave aromatic and hydrophobic amino acid residues at the P1 position. Keratins are composed of 50 to 60% aromatic and hydrophobic residues, which could partially explain the keratinase specificity for keratin [120–122]. Most of these peptidases are stabilized by divalent cations like $Ca^{2+}$ and are extracellular [119,123,124].

## 5.2. Problems Associated with Keratinase Assays

There are several issues with the current methods used to characterize keratinases. The assays are not standardized in the literature in terms of reaction conditions and substrates. The most common method used to measure keratinase activity is a colorimetric assay that uses the commercially available derivative of wool, keratin azure [125] or azokeratin (sulfanilic acid-azokeratin [126]). However, batch variability and the fact that the chromogenic agents are only bound to the outer portion of the substrate compromises reproducibility. Quantification of the soluble peptides generated by hydrolysis of keratin has also been used to determine the effectiveness of keratinases on keratin substrates. Common quantification methods used are Bradford [95,127], Lowry [128,129] or measurement of absorption at 280 nm [106,111] (see Table 4). Each of these methods have several limitations. The Coomassie Blue dye used in the Bradford method preferentially reacts with arginine and lysine in the peptide mix and alkaline pH and detergents interfere with the reaction. The Folin–Ciocalteu dye used in the Lowry method oxidizes the aromatic amino acids residues in the protein and is affected by reducing agents. Only tyrosine, tryptophan and cysteine absorb at 280 nm and other compounds like DNA in the solution can interfere with the measurement [130]. The simplest and probably most accurate method for quantifying keratinase activity is the measurement of weight loss when the insoluble keratin substrates like mammalian hair fibres, feathers or wool are solubilised through hydrolysis [96,127].

Table 4 describes a variety of substrates that have been used to assay keratinase activity in the literature. The substrates that were used include keratin azure (wool), keratin powder, soluble keratin, keratin (undefined), feathers and feather meal powder. It was not possible to ascertain the source and integrity of most of these keratin substrates from the papers. The pretreatment of these substrates is also an important aspect in determining the keratinase activity. Keratin powder and solubilized keratin were generally obtained from commercial sources; however, the sources and preparation were not described. Pretreatments like autoclaving and milling [103,107], or treating with solvents at high temperatures [106], are known methods for keratin powder preparation from the literature. In the case of the $rK_{27}$ keratinase, the feather powder used in the assay was autoclaved and dried at 60 °C [103]. These preparation methods, as already described in Section 3, would compromise the keratin structure. The keratinases, KerRP [96], Ker-A1 [97] and SAPB [102] were assayed on keratins of unknown source. In the WF146 protease assay, the feather substrate was washed with ethanol prior to use in the assay, which would likely remove the protective lipid layer [108].

Co-treatment can also affect the integrity of the keratin substrates during enzymatic hydrolysis [125,131,132]. Except for KerQ7 [104], all assays in Table 4 were carried under alkaline conditions between pH 8 and pH 12.5 and temperatures ranging from 50 to 80 °C. These conditions would most likely contribute to the weakening the keratin structure. Keratinase assays with SAPDZ [100], KerQ7 [104], KERDZ [94], and KERAK-29 [95] were supplemented with the divalent cations $Ca^{2+}$ or $Mn2^+$. Divalent cations are known to stabilize serine proteases [114,117]. Other keratinase studies added reducing agents, like β-mercaptoethanol (protease C2 [106], WF146 protease [108]) or dithiothreitol (SFP2 [99]) to the assay mixture. Reducing agents are known to break the disulphide bond leading to disruption of the keratin.

**Table 4.** Keratinase pH and temperature optimums of the selected S1, S8 and M4 keratinases with associated assay conditions. See text for further details of the assays.

| Protein | * pH | * Temp (°C) | Conditions | PT | CT |
|---|---|---|---|---|---|
| | | | **S1A Peptidases** | | |
| KERDZ | 11 | 80 | 10 g/L keratin azure, 50 mM bicarbonate-NaOH buffer, pH 11 mixed 1:1 with the enzyme, 30 min, 80 °C, 200 rpm ($Abs_{595nm}$). | - | 2 mM $CaCl_2$ |
| KERAK-29 | 10 | 70 | 1 mL of 10 g/L keratin azure, 100 mM Glycine-NaOH buffer mixed 1:1 with the enzyme, pH 10, 20 min, 70 °C ($Abs_{595nm}$). | - | 5 mM $MnSO_4$ |
| SFP2 | 10 | 60 | 5 mg keratin azure, 50 mM Tris-HCl, pH 8.5, 1 h, 37 °C ($Abs_{595nm}$). | - | 10 mM DTT |
| NAPase | 12.5 | 60 | 60 mg wool keratin powder, Glycine-NaOH, pH 10 or 50 mM KCl-NaOH, pH 12.5, 30 °C, 2 h ($Abs_{280nm}$). | Not specified | - |
| | | | **S8A Peptidases** | | |
| SAPDZ | 12.5 | 85 | 10 g/L keratin azure, 100 mM KClNaOH, 250 rpm, 20 min incubation, 85 °C ($Abs_{595nm}$). | - | 5 mM $CaCl_2$ |
| KerRP | 9 (11) | 60 (65–70) | 0.8% w/v keratin diluted 1:1 in enzyme, 1 h incubation, 60 °C ($Abs_{280nm}$). | Not specified | - |
| KerSMD | 8 | 60 | 1% w/v soluble keratin, 50 mM Gly-NaOH, 20 min, 50 °C (Folin–Ciocalteu). | Not specified | - |
| KerSMF | 9 | 60 | 1% w/v soluble keratin, 50 mM Gly-NaOH, 20 min, 50 °C (Folin–Ciocalteu). | Not specified | - |
| KerA1 | 9 (10) | 60 (65) | 0.8% w/v keratin diluted 1:1 in enzyme solution, 1h, 50 °C ($Abs_{280nm}$). | Not specified | - |
| SAPB | 10.6 | 65 | 1% keratin w/v, on 100 mM glycine-NaOH Buffer, pH 10.6, 30 min, 55 °C. 2 mM $CaCl_2$ supplemented. | Not specified | - |
| rK27 | 9 | 70 | 20 mg feather powder, Gly-NaOH 50 mM, 1 h ($Abs_{280nm}$). | Washed with Triton X-100 (1%), rinsed with water, autoclaved, dried in an oven at 60 °C for 1 h, milled then sieved with 2 mm pore size. | - |
| KerQ7 | 7 | 30 | 10 g/L keratin azure, 50 mM HEPES buffer, 30 min, 200 rpm ($Abs_{595nm}$). | - | 1 mM $CaCl_2$ |
| Vpr | 8.5 | 50 | 2% w/v chopped feather keratin, 50 °C, 15 min, pH 7.5. | - | - |
| KBALT | 8 | 85 | 5 mg keratin azure, 0.8 mL buffer, 15 min incubation, pH 6 to 12, 25 to 95 °C ($Abs_{595nm}$). | - | - |
| YT06 protease | 8–9 | 65 | 1% soluble keratin, 50 mM Gly-NaOH, pH 9, 20 min (Folin–Ciocalteu). | Not specified | - |
| Protease C2 | 11 | 60–80 | 5% keratin powder, 50 mM Tris-HCl pH 8, 60 min 60 °C ($Abs_{280nm}$). | 100 °C incubation in DMSO for 2 h. Protein precipitated with acetone 2:1 v/v [133] | 0.5% β-ME |
| rMtaKer | 10 | 65 | 1% feather powder on 50 mM HEPES, pH8.0, 25–95 °C. Supplemented with 10 mM $CaCl_2$, 150 mM NaCl (Ninhydrin). | Chicken feathers rinsed, air-dried, ground by ball mill. | - |
| WF146 protease | - | 80 | 10 mg of feathers, 50 °C or 80 °C, 1 ml Tris-HCl 50 mM buffer, pH 8.0, 10 mM $CaCl_2$, multiple time points from 0 to 20 h ($Abs_{280nm}$). | 70 Ethanol wash, rinse water, dry, cut 2–3 mm long | 1% β-ME |
| | | | **M4 Peptidases** | | |
| RecGEOker | 9 | 60 | 4 mg keratin azure, 50 mM Tris-HCl, pH 7.8, 1 h (Wool-Folin–Ciocalteu; $Abs_{595nm}$). | - | - |
| KerP | 9 | 50 | 20 mg chicken feathers, Glycine-NaOH buffer, pH 10, 60 °C, 60 min ($Abs_{280nm}$) | - | - |

Note: Source organism, accession numbers and references can be found in Table 3; * in some cases pH and temperature optimums were determined on both casein and keratin substrates. The casein optimums are in brackets; Temp = temperature; PT = pretreatment; CT = co-treatment; β-ME = β-mercaptoethanol; DTT = dithiothreitol. Quantification methods, where available, are in brackets after the assay condition description.

The challenges with the keratinase assays discussed above highlight the need for standardized assays and substrates used to test keratinases and the challenges faced in attempting to compare and analyse data from the literature when the assays are not comparable.

## 5.3. The Effect of Additives on Selected S1, S8 and M4 Keratinases

Various additives were tested on the selected S1, S8 and M4 keratinases-cationic, anionic and neutral detergents, oxidizing agents, reducing agents, mono- and divalent metals. Table 5 contains a summary of additives that had a positive effect on keratinase activity. A positive effect was defined as ≥ 110% activity compared to the control without additive. Some of the papers used keratin as the assay substrate, some used casein and in some cases, both were tested.

**Table 5.** Influence of additives on the activity of selected S1, S8 and M4 keratinases. Numbers in brackets correspond to the % activity compared to 100% in the absence of the additive.

| Protein | Metal ions (%) | | Detergents (%) | | Reducing Agents (%) | Solvents/Others (%) |
|---|---|---|---|---|---|---|
| **S1A Peptidases** | | | | | | |
| KERDZ | $Ca^{2+}$ (270) $Mg^{2+}$ (180) $Fe^{2+}$ (145) | | | | | |
| KERAK-29 | $Ca^{2+}$ (150) $Mg^{2+}$ (110) $Mn^{2+}$ (210) | | Zwittergent (114) Tween-20 (130) Triton X-100 (132) Tween-80 (145) TTAB (116) CHAPS (140) | Sulfobetaine (135) LAS (118) SDS (115) CTAB (110) | β-ME (102) | $H_2O_2$ (170) |
| SFP2 | $Cu^{2+}$ (149) $Ni^{2+}$ (116) | | | | DTT (278) β-ME (235) | |
| NAPase | | | | | | **Isopropanol (130)** |
| **S8A Peptidases** | | | | | | |
| SAPDZ | $Ca^{2+}$ (450) $Mg^{2+}$ (195) $Mn^{2+}$ (280) | $Zn^{2+}$ (180) $Cu^{2+}$ (110) $Co^{2+}$ (113) | | | | |
| KerRP | * $Ca^{2+}$ | | | | | |
| KerSMD | $Ca^{2+}$ (112) | | | | $Na_2SO_3$ (116) | |
| KerSMF | ** $Ca^{2+}$ (105) | | Tween-20 (112) | | $Na_2SO_3$ (115) DTT (115) | |
| KerA1 | $Ca^{2+}$ (123) $Mg^{2+}$ (199) $Na^+$ (135) | | Tween 80 (113) | | β-ME (Casein 100) (Keratin 192) | |
| SAPB | $Ca^{2+}$ (157) $Mg^{2+}$ (112) $Na^+$ (118) | | LAS (114) Tween 80 (119) | Tween 20 (117) SDS (119) | β-ME (192) | Urea (165) $H_2O_2$ (168) |
| rK27 | Stability only tested | | Triton X-100 (677) Tween-80 (242) Saponin (461) Sodium Cholate (276) SDS (186) | | DTT (267) β-ME (323) | NaClO (276) $H_2O_2$ (275) |
| KerQ7 | $Ca^{2+}$ (417) $Mg^{2+}$ (175) $Mn^{2+}$ (250) | $Ba^{2+}$ (121) $Sn^{2+}$ (115) | | | | |
| KBALT | $Ca^{2+}$ (127) $Mg^{2+}$ (134) | $Zn^{2+}$ (129) $Ba^{2+}$ (115) | SDS (128) | | β-ME (102.5) | |
| YT06 Protease | $Mg^{2+}$ (118) $Mn^{2+}$ (196) | $Ni^{2+}$ (120) $Ba^{2+}$ (115) | Tween-20 (170) | | β-ME (623) | |
| **M4 Peptidases** | | | | | | |
| RecGEOker | $Mg^{2+}$ (112) $Mn^{2+}$ (116) $Zn^{2+}$ 1 mM (58); 10 mM (52) $Ca^{2+}$ 1 mM (101); 10 mM (66) | | Triton X-100 (115) Tween 40 (180) Tween 60 (133) | Tween 80 (122) Triton X-305 (153) | DTT (139) | |

Note: Source organism, accession numbers and references can be found in Table 3; Bold = tested on a keratinous substrate; Not bold = tested on casein; * = Only tested for binding; ** included for comparison with KerSMD; underlined = denaturing detergents; DTT = dithiothreitol; β-ME = β-mercaptoethanol; LAS = linear alkylbenzene sulfonate; SDS = sodium dodecyl sulfate; TAED = tetraacetylethylenediamine; TTAB = tetradecyltrimethylammonium bromide; CHAPS = 3-[(3-cholamidopropyl)dimethylammonio]-1-propanesulfonate; CTAB = cetrimonium bromide.

Despite there being large differences in concentrations of metals additives, incubation time and temperature, in general, supplementation with $Ca^{2+}$ showed the largest increase in activity except for KerSMF [101] and kerA1 [97]. In the case of KerSMF, $Ca^{2+}$ had no effect on activity and in the case of kerA1, $Mg^{2+}$ addition increased activity by 199% compared to 123% for $Ca^{2+}$. In general, $Ca^{2+}$ > $Mg^{2+}$ > $Mn^{2+}$ had a positive effect on all the S1 and S8 keratinases (Table 5). The effect of these divalent metals on M4 metalloproteases is discrete compared to serine proteases. Only the addition of magnesium and manganese ions resulted in keratinase activity slightly above the control without additives. These divalent ions have been described to stabilize the active structure of the peptidases by binding to the metal-binding sites [100]. Other explanations for the higher activity are possible stabilization of enzyme/substrate complex [101] or formation of salt or ion bridges that maintain the enzyme conformation [97,122,128]. Furthermore, these metal ions reduce thermal denaturation [134]. Lin et al. observed that aqualysin, a thermostable peptidase from the S8 family, was only stable at high temperatures in the presence of 1 mM $Ca^{2+}$ [135].

Several studies were carried out on the keratinase activity in the presence of metal ions ($Zn^{2+}$, $Cu^{2+}$, $Co^{2+}$, $Ba^{2+}$, $Sn^{2+}$ and $Ni^{2+}$) were carried out (Table 5). The addition of the metal ions improved activity between 10% and 29% except for SAPDZ [100], where $Zn^{2+}$ addition increased activity by 80% and $Cu^{2+}$ addition increased activity of SFP2 [99] by 49%. Li et al. characterized SFP1, a non-keratinolytic peptidase similar to SFP2 and produced by the same organism [99]. It showed increased activity with copper ions, possibly due to the stabilization of the enzyme. Copper ions acting as a stabilizer has rarely been described in previous serine protease studies, and it is not known whether there is a copper-binding site stabilizing the enzyme [136]. In another example, peptidases were more stable in the presence of copper ions, which resulted in a reduction in both autolysis and thermal inactivation rates [137].

Detergents, in general, enable the disruption or formation of hydrophobic and hydrophilic bonds and assist in the extraction of proteins into aqueous media [138]. Detergents can act as denaturing agents on enzymes. Denaturing detergents are anionic (SDS, LAS) or cationic (CTAB, TTAB). They denature proteins by breaking protein–protein interactions. Non-denaturing detergents are non-ionic (Triton X-100, Tweens, cholate, saponin) or zwitterionic (CHAPS, sulfobetaine, zwittergent), and their action is milder and enzyme function is usually maintained. In most cases the addition of denaturing and non-denaturing detergents resulted in an increase in activity (110–150%). However, the addition of the non-ionic detergents to the assay mixture with keratin as substrate of $rK_{27}$ had a dramatic effect on activity compared to the control without detergent [103]. Activities of 677% (Triton X-100), 242% (Tween 80), 461% (saponin) and 276% (cholate) were achieved. The addition of the anionic denaturing detergent, SDS to the assay increased the keratinase activity to 186%. The addition of the non-ionic detergents, Tween 40, Tween 60, Tween 80 and Triton X-305 to the assay mixture with keratin as substrate for the M4 keratinase, RecGEOker [109], showed increased activity to 180%, 133%, 122% and 153%, respectively. This example showed a definite trend of increasing activity with decreasing Tween 80 (monounsaturated C18 derivative) < Tween 60 (saturated C18 derivative) < Tween 40 (saturated C16 derivative). The partial solubilizing action of detergents on the insoluble keratin substrate might explain why both denaturing and non-denaturing detergents have a positive effect on keratin hydrolysis. There are insufficient examples to confirm this Tween effect on keratinases in general.

The reduction of disulphide bonds, destabilizes keratins and acts synergistically with keratin hydrolysis in nature [6,8–10]. Sodium sulphite, dithiothreitol (DTT) and β-mercaptoethanol were tested on some of the keratinases in Table 5. The reducing agents had a positive effect on all S8 keratinases tested with keratin as the substrate. The increase in activity ranged from 115% for $Na_2SO_3$ (KerSMF [101]) to 623% for β-mercaptoethanol (YT06 protease [98]) except in the case of KBALT [63], where β-mercaptoethanol had no effect on the activity. β-Mercaptoethanol doubled the activity of SAPB [102] when tested with casein as substrate. DTT also increased the activity of the M4 keratinase, RecGEOker (139% [109]), when tested with keratin as substrate. None of the S1 enzymes were tested

with keratin and reducing agents. However, the two assays with casein and reducing agent showed on one hand, no effect from β-mercaptoethanol on KERAK-29, [95] and on the other hand, a considerable effect on SFP2 (DTT, 278%; β-mercaptoethanol, 235% [99]). It should be noted that where disulphide bonds present in the enzyme are essential for function the inclusion of reducing agents may negatively affect activity.

Chaotropic agents are comparable to detergents, breaking non-covalent interactions and allowing protein denaturation [139–141]. Urea and isopropanol are chaotropic agents (Table 5). The activity of SAPB [102] was increased to 165% in the presence of urea compared to the control and the activity of NAPase [111] was increased to 130% in the presence of isopropanol [111].

The effect of the oxidizing agents, $H_2O_2$ and sodium hypochlorite, was also studied on three S8 and S1 keratinases, SAPB [102], rK$_{27}$ [103] and KERAK-29 [95]. Activity was significantly increased in all cases (Table 5).

In most cases the effect of additives like divalent cations, detergents, reducing agents, chaotropic agents and oxidizing agents have a positive effect on keratinase activity. Nearly all compounds capable of disrupting the integrity of the keratin structure without inactivating the keratinase appear to have a positive effect on keratinase activity. The effect of compounds disrupting the keratin structure was, in some cases like rK$_{27}$ [103], significant.

## 5.4. Substrate Specificity

Table 6 summarizes the substrate specificity data of the selected keratinases from the literature. In general, a variety of keratins and other proteins like gelatin, casein and albumin were tested. To compare the selected keratinases, the values in Table 6 have been normalized using the activity of designated keratin substrates (keratin azure, keratin, feather or wool) as 100% activity.

KerQ7 [104] was the only keratinase in Table 6 tested on multiple types of keratins. KerQ7 showed a preference for the β-keratin-rich feather meal and feathers. The activity on feather meal was only 16% higher than feathers. The activity on rabbit hair, goat hair and bovine hair was 88%, 74% and 50% of the activity on feathers, respectively, whereas activity on wool was only 12%. These substrates are rich in α-keratins [4,9,12,13]. Substrate fibre thickness and fibre surface area may also contribute to the variations in enzyme activity. Nonetheless, the keratinase activity toward various substrates is likely to be multifactorial. KerSMD and KerSMF, from *Stenotrophomonas maltophilia*, showed less activity towards feather powder and wool than keratin azure [101]. KerSMD and KerSMF had similar activity on feather powder (54% and 71%, respectively) and wool (59% and 78%, respectively). However, KerSMD showed an activity of 1589% towards soluble keratin compared to an activity of 126% for KerSMF on the same substrate.

No trends were observable on non-keratin substrates. For example, SAPDZ [100] showed 81% activity on gelatin compared to keratin, whereas the activity of kerA1 [97] and SAPB [102] on gelatin was 22% and 146% compared to keratin, respectively. The same inconsistencies can be seen with casein. The activities of SAPB [102] and KerSMD [101] on casein are 153% and 2800%, respectively, compared to keratin azure, whereas KerSMF [101] has only slightly lower activity on casein (91%) compared to keratin azure.

Keratinases are known for their activity on "hard-to-degrade" proteinogenic substrates. Most of the characterized keratinases in the literature are also capable of degrading collagen, which is an example of another complex and hard-to-degrade substrate [142]. A study in 2008 characterized the first keratinase without collagenase activity [143]. Only three enzymes from the S8 family were tested on collagen or azocoll (azocollagen). Vpr [105] presented collagenase activity (129%) and while SAPDZ [100] did not. Protease C2 [106] showed a surprisingly high activity (24000%) on azocoll compared to keratin azure (100%). KERDZ [94], from the S1 peptidase family, had no activity on collagen, whereas RecGEOker [109], belonging to the M4 metallopeptidase family, was able to hydrolyse collagen. The differences in activity between substrates may be attributed to the specific peptide sequences in the substrates and the sequence specificity of the enzymes.



**Table 6.** Substrate specificity of the selected S1, S8 and M4 keratinases. Numbers in brackets correspond to the % activity relative to other substrates.

| Protein | Keratins (%) | | Natural Proteins (%) | | Modified Protein (%) | Esters and Others (%) | |
|---|---|---|---|---|---|---|---|
| **S1A Peptidases** | | | | | | | |
| KERDZ | Keratin (100) [2] | | Gelatin (90) Casein (79) Albumin (75) Elastin (50) | Myoglobin (41) Hemoglobin (20) Collagen type 1/2 (0) | Azocasein (80) Azoalbumin (70) | BAEE (91) TAME (100) BCEE (95) BTEE (0) ATEE (0) | |
| SFP2 | Keratin (100)[2] | | Casein (111) | | | | |
| **S8A Peptidases** | | | | | | | |
| SAPDZ | Keratin (100) | | Gelatin (81) Casein (95) Albumin (72) | Hemoglobin (66) Collagen type 1/2 (0) | Azocasein (91) Keratin Azure (100) [1] | BAEE (0) BCEE (0) | BTEE (100) ATEE (95) |
| KerSMD | Feather powder (54) Soluble keratin (1589) Wool (59) | | Casein (2800) | | Keratin Azure (100) [1] | | |
| KerSMF | Feather powder (71) Soluble keratin (126) Wool (78) | | Casein (92) | | Keratin Azure (100) [1] | | |
| KerA1 | Keratin (100) [2] | | Gelatin (22) Casein (222) | Elastin (54) BSA (97) Egg albumin (4) | Azocasein (177) Azokeratin (92) | | |
| SAPB | Keratin (100)[2] | | Gelatin (146) Casein (153) | Bovine serum albumin (80) Egg albumin (18) Gluten (30) | Azocasein (123) Azokeratin (96) | BTEE (109) ATEE (115) | |
| rK$_{27}$ | Powdered chicken feather > haemoglobin > meat protein > hoof keratin > fibrin > elastin > gelatine > casein > BSA > azocasein > keratin azure | | | | | | |
| KerQ7 | Rabbit hair (88) Goat hair (74) Bovine hair (50) | Wool (12) Feather meal (116) Feather (100) [3] | | | | | |
| Vpr | Feather meal (50) Keratin (100) [2] | | Gelatin (147) Casein (156) | Fibrin (145) Collagen (129) | | | |
| Protease C2 | Bovine hair (274) Feather (439) | | Albumin (8571) Elastin (11) | | Keratin azure (100) [1] Azocasein (102857) Azocoll (24000) | | |
| **M4 Peptidases** | | | | | | | |
| RecGEOker | Wool (100) [5] | | Gelatin (92) Casein (95) | Albumin (37) Collagen type 1 (98) | | | |

Note: Source organism, accession numbers and references can be found in Table 3; Activity normalized to the following substrates—[1] keratin azure, [2] keratin; [3] feather, [4] wool; BAEE = N-α-benzoyl-L-arginine ethyl ester; TAME = N-α-p-tosyl-L-arginine methyl; BCEE = benzoyl-citrulline ethyl ester; BTEE = N-benzoyl-L-tyrosine ethyl ester; ATEE = N-acetyl-L-tyrosine ethyl ester; DTNB = 5,5′-dithiobis-(2-nitrobenzoic acid); Azocoll = commercially available azocollagen.

Some enzymes also showed esterase activity, which may be of importance for facilitating enzyme access to the substrate. Fatty acids of the cuticle surface are linked via a thioester to the protein matrix below in keratin fibres and feathers [14]. Only three enzymes in Table 6 have been characterized on ester substrates. The two S8 family peptidases—SAPDZ [100] and KerRP [96]—appear to have similar ester substrate affinity with both showing activity against N-benzoyl-L-tyrosine ethyl ester (BTEE) and N-acetyl-L-tyrosine ethyl ester (ATEE). In contrast, KERDZ (S1 family) had no activity towards these substrates but was active towards N-α-benzoyl-L-arginine ethyl ester (BAEE), N-α-p-tosyl-L-arginine methyl (TAME) and benzoyl-citrulline ethyl ester (BCEE) [94].

In general, the peptidases from S1, S8 and M4 families (Table 6) present varied substrate specificities. There are limited examples in the S1 and M4 families to detect trends but even within the S8 peptidases examples there were no obvious substrate preferences.

## 6. Potential Applications of Keratinases

New keratinases with improved properties for commercialization and the keratin hydrolysates they produce represent an opportunity for adding value to keratin waste.

Commercial keratinases are sold for a variety of applications (Table 7) such as the degradation of infectious prions, as supplements for animal feed to improve its nutritional value, removal of corns and calluses from skin, treatment of acne and nail fungi and, they are also incorporated into cosmetic skin peeling and depilatory creams [6,52,144]. Other applications include the use in cleaning products for unblocking drainpipes and septic tanks.

**Table 7.** Some examples of commercial keratinases.

| Trade Name | Source | EC Number | Substrate or Function | Supplier |
|---|---|---|---|---|
| Versazyme [1,3] | *Bacillus licheniformis* | 3.4.21.62/ S8 family | Improving nutritional value of poultry feed & prions degradation | Bioresource Int'l, Inc. |
| Ronozyme ProAct [2] | *Nocardiopsis prasina* | 3.4.21.-/serine protease | Improving nutritional value animal feed | DSM/Novozymes |
| Cibenza DP100 [2] | *Bacillus licheniformis* PWD-1 | - | Improving nutritional value animal feed | Novus International |
| Pure Keratinase 100 [3] | *Bacillus licheniformis* PWD-1 | - | Prion degradation from medical & dental instruments | Proteus Biotech |
| BioGuard Plus [3] | Proprietary blend of microorganisms – incl. keratinase producer | - | Cleaning drainpipes, septic tanks & digesters | RuShay Inc. |
| Keratoclean sensitive PB [3] | *Bacillus licheniformis* (PB333 keratinase) | - | Treatment acne, dead skin removal, promotes cell renewal | Proteus Biotech |
| Keratoclean Hydra PB [3] | *Bacillus licheniformis* | - | Removal of corns & call uses, acne, Hirsutism, peeling | Proteus Biotech |
| FixaFungus [3] | - | - | Treatment of toenail fungal infections | Proteus Biotech |

Source: [1] [6], [2] [127], [3] [52].

There are also a number of promising applications of keratinases that have not been commercialized to date: dag or manure balls removal from cattle hides and tails [145]; extraction of glucocorticoids from chicken feathers to monitor the stress level in poultry breeding and production programmes [146] extraction of chicken feather cholesterol as a precursor to bile salts that can be used to produce bio-emulsifiers and biosurfactants in the cosmetic industry [18]; selective hydrolysis of wool from wool/polyester or mixed textiles to facilitate textile recycling [147]; and dehairing of hides in the leather industry [64,65,82].

The use of keratinases for the processing of keratinous waste might be advantageous for high value products. The use of enzymes instead of thermochemical methods for keratin hydrolysis reduces chemical modification arising from harsh chemical hydrolysis and might allow a degree of control of the peptide composition that is produced. Keratin hydrolysates are widely used in protein feed supplements [18]. Feather waste, for example, is hydrolysed with saturated steam under high pressure (sometimes with the addition of lime) to produce feather meal, which is used as a feed supplement [148]. These conditions lead to the loss or modification of some of the amino acids, which impacts the nutritional value and digestibility of the feather meal. Hydrolysis with keratinases might offer an alternative, which reduces the energy requirements of the process and enhances the nutritional value of the supplement.

Keratin-derived bioactive peptides have been reported in the literature. These peptides have a range of activities like antimicrobial [149], antihypertensive [150], anti-inflammatory [151–154], antioxidant [149,150,155], inhibition of early stage amyloid aggregation [156], antidiabetic [157] or anti-aging [158–160] depending on the keratin source and the method of preparation. Producing protein feed supplements with antioxidant or anti-inflammatory properties as well as skin and hair

products with antioxidant, anti-inflammatory, antimicrobial or anti-aging properties would most likely increase the value of these products.

Keratin peptides and subunits can spontaneously self-assemble [161]. This property can be exploited to form biomaterials like hydrogels, films, sponges, scaffolds and nanofibres for tissue engineering, wound healing, fibroblast cultivation and treatment of burns [161–164]. The production of smart biocomposites is also of interest. An example is the production of transparent plastic film containing citric acid from wool hydrolysate [165]. The plastic has excellent biocidal activity and could be used as a functional packaging for food.

The examples described above demonstrate the commercial potential of keratinases and the large number of opportunities they offer for adding value to keratin waste by producing bioactive protein feed supplements, personal care products and biomaterials from keratin hydrolysates.

## 7. Discovery and Future Research

The standout problem with the characterization of keratinases, demonstrated by the analysis of the assay conditions in this review, is lack of standardization of the keratinase assay combined with the small number of sequenced peptidases with keratinolytic activity that have been biochemically characterized. Both of these issues hamper the identification and comparison of true keratinases. Current experimental conditions vary in temperature, pH, buffer types and concentration, additives, substrates and their pretreatment biasing possible conclusions. It is unclear whether some proteases are keratinases or whether pretreatment or co-treatment influences their keratinolytic activity to some degree.

The uncertainty in defining keratinases and highly variable characterization of keratinases in the literature increases the challenge of finding new keratinases based on literature data or from sequence databases. However, the discovery of new keratinases is critical for expanding the opportunities for waste keratin valorisation. It would be desirable to identify new keratinases with high activities and specificities enabling control over cleavage sites, peptide molecular weights and amino acid side chain modifications.

Standardized experiments combined with phylogenetic studies and sequence analyses are needed. Standardized experiments, which avoid pre- or co-treatments, would determine the true protease activity on keratin substrates and reduce possible experimental biases. An in-depth phylogenetic analysis would help to clarify the position of keratinases within the phylogenetic trees of the peptidase families in which they are found and may help focus the search for new peptidases with keratinolytic activity. A comprehensive sequence analyses, aimed at the identification of conserved sites between peptidases with keratinolytic activity, as well as the presence of specific domains that possibly contribute to their ability to hydrolyse keratin, may assist in the development of algorithms to search the vast sequence space of the peptidase families.

**Author Contributions:** Conceptualization, R.S., K.R., J.P.D.O.M., L.N., M.N., G.C. and Z.Z.; investigation, J.P.D.O.M. and K.R.; writing—original draft preparation, J.P.D.O.M. and K.R.; writing—review and editing, R.S., L.N., M.N. and G.C.; supervision, R.S.; project administration, R.S.; funding acquisition, R.S. All authors have read and agreed to the published version of the manuscript.

**Funding:** This research was funded by the Australian Government Department of Agriculture, grant number RnD4Profit-16-03-002.

**Acknowledgments:** This project is supported by Meat and Livestock Australia through funding from the Australian Government Department of Agriculture as part of its Rural R&D for Profit program and the partners.

**Conflicts of Interest:** The authors declare no conflicts of interest. This manuscript was approved for publishing by Meat and Livestock Australia and the Australian Government Department of Agriculture.

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
