# Peer review of "Challenges and Opportunities in Identifying and Characterising Keratinases for Value-Added Peptide Production"

_catalysts, doi:10.3390/catal10020184_

Round 1

Reviewer 1 Report

This article is devoted to a review of existing problems associated with enzymatic hydrolysis of such a difficult substrate as keratin. The review is very well written, well arranged and full of valuable information about proteinases with a specific activities towards keratins. I have no complaints about the scientific content, nor the organization of the review, nor the English style. The only recommendation I would like to give is to add two figures: 1. keratin structures (in my opinion, this figure is necessary in the section 2. Keratin, a complex and strong structure); 2. a schematic representation of the enzymatic keratin hydrolysis in the section 4. Microbial degradation of keratin.

Author Response

Point 1: The only recommendation I would like to give is to add two figures: 1. keratin structures (in my opinion, this figure is necessary in the section 2. Keratin, a complex and strong structure); 2. a schematic representation of the enzymatic keratin hydrolysis in the section 4. Microbial degradation of keratin.

Response 1: We have provided the figures as requested. The first figure is adapted from another paper under a creative commons licence (https://creativecommons.org/licenses/by/4.0/deed.en). The second figure we generated ourselves.

Reviewer 2 Report

This review covers a field of great interest to Catalysts readers in the field of Biocatalysis. However, a few points deserve attention for further publication in Catalysts journal. I suggest that it is accepted for publication after the following revisions:

I suggest the authors add a figure of the keratin structure in section 2.

I suggest the authors add a point about the heterologous expression of keratinases.

What about amino acid sequence homology between different keratinases? There is barely information about it in the article.

Table 4 needs to be improved. It would be interesting to add a column with the corresponding reference. How is keratinase activity quantified in these assays?

The authors should better explain in the legend of table 5 that it shows the influence of different additives on the hydrolysis of keratin. Clarify in the legend the meaning of the numerical data in brackets (increased keratinase activity).

A red dot appears on line 373.

Author Response

Point 1: I suggest the authors add a figure of the keratin structure in section 2.

Response 1: We have added the figure as requested.

Point 2: I suggest the authors add a point about the heterologous expression of keratinases.

Response 2: We have added a paragraph on heterologous expression in Section 5.

Point 3: What about amino acid sequence homology between different keratinases? There is barely information about it in the article.

Response 3: We have added information about available sequence homology in Section 5.

Point 4: Table 4 needs to be improved. It would be interesting to add a column with the corresponding reference. How is keratinase activity quantified in these assays?

Response 4: It is mentioned in the notes under Table 4 that all relevant references can be found in Table 3 so we do not believe it is necessary to repeat this information in another column in Table 4. We have added how keratinase activity is quantified in the assays in brackets in the 'Conditions' column in Table 4. 

Point 5: The authors should better explain in the legend of table 5 that it shows the influence of different additives on the hydrolysis of keratin. Clarify in the legend the meaning of the numerical data in brackets (increased keratinase activity).

Response 5: We have changed the figure legend as suggested. We also make a similar change to the legend for Table 6 following this good suggestion.

Point 6: A red dot appears on line 373.

Response 6: The red dot has been removed.